# Unsupervised Learning for Optimal Transport plan prediction between unbalanced graphs

**Sonia Mazelet**
CMAP, Ecole Polytechnique
Palaiseau, France
sonia.mazelet@polytechnique.edu

**Rémi Flamary**
CMAP, Ecole Polytechnique
Palaiseau, France
remi.flamary@polytechnique.edu

**Bertrand Thirion**
Mind, Inria-Saclay
Palaiseau, France
bertrand.thirion@inria.fr

## Abstract

Optimal transport between graphs, based on Gromov-Wasserstein and other extensions, is a powerful tool for comparing and aligning graph structures. However, solving the associated non-convex optimization problems is computationally expensive, which limits the scalability of these methods to large graphs. In this work, we present Unbalanced Learning of Optimal Transport (ULOT), a deep learning method that predicts optimal transport plans between two graphs. Our method is trained by minimizing the fused unbalanced Gromov-Wasserstein (FUGW) loss. We propose a novel neural architecture with cross-attention that is conditioned on the FUGW tradeoff hyperparameters. We evaluate ULOT on synthetic stochastic block model (SBM) graphs and on real cortical surface data obtained from fMRI. ULOT predicts transport plans with competitive loss up to two orders of magnitude faster than classical solvers. Furthermore, the predicted plan can be used as a warm start for classical solvers to accelerate their convergence. Finally, the predicted transport plan is fully differentiable with respect to the graph inputs and FUGW hyperparameters, enabling the optimization of functionals of the ULOT plan.

## 1 Introduction

**Graph alignment**   In many graph data applications, aligning or matching nodes between two graphs is necessary. Examples include object detection, where semantic correspondences between objects in two images from different domains can make the model more adaptive [12]; graph edit distance [22, 18], where the goal is to compute the distance between two graphs, which requires finding the best correspondence between their nodes; shape matching [20], or brain alignment across subjects [30].

But the problem of graph matching is challenging because of the combinatorial nature of the problem, which can often be reformulated as a Quadratic Assignment Problem (QAP) known to be NP-hard [16]. One strategy that has been proposed recently is to use deep learning to learn the matching in a supervised setting [36], [25]. These methods typically rely on a graph neural network (GNN) to learn a representation of the nodes and then use a matching algorithm to compute the correspondence between the nodes. However, the problem is even more difficult when the graphs are unbalanced, i.e. when they have different numbers of nodes or when some nodes have noisy features or connections.

39th Conference on Neural Information Processing Systems (NeurIPS 2025).

**Optimal transport between graphs**   In recent years, Optimal Transport (OT) has emerged as a powerful tool for solving the graph matching problem. It can be seen as a continuous relaxation of the QAP with the Gromov-Wasserstein (GW) distance [17, 21, 40], which is a generalization of the classical Wasserstein distance to distributions in different metric spaces. Extensions of the GW distance to labeled graphs have been proposed, such as the Fused Gromov-Wasserstein (FGW) distance [31]. But those OT problems put strong constraints on the transport plan, which makes them very sensitive to noise, outliers and local deformations. This is why Unbalanced GW [27, 5] and Fused Unbalanced GW (FUGW) [30] were proposed to generalize the GW and FGW distances to unbalanced settings with application in positive unlabeled learning and brain alignment.

**Complexity of classical optimal transport solvers**   Quadratic OT problems such as FGW and FUGW are non-convex. Classical solvers rely on a block coordinate descent algorithm to iteratively solve linearized versions of the problem. This linearization is of complexity $O(n_1 n_2^2 + n_2^2 n_1)$, where $n_1$ and $n_2$ are the number of nodes in the two graphs [21]. This makes the method unscalable for large graphs, for applications where we need to compute the transport plan for many pairs of graphs or when validation of the hyperparameters is necessary. This is especially problematic in the unbalanced case where the solution is very sensitive to the parameters.
The ML community has recently proposed to use deep learning for accelerating or solving OT problems. For example [11] proposed to estimate the OT mapping using a neural network and extensions to the GW have been proposed in [19, 38]. But the most relevant work for our purpose is Meta OT [2] which proposed to learn to predict the dual potentials of the entropic OT problem with a neural network. But this approach is limited to classical entropy regularized OT and not directly applicable to the quadratic FUGW problem which motivates our proposed approach detailed below.

**Contributions**   We propose in this paper a new method to learn a neural network that predicts the transport plan of the FUGW problem denoted as Unsupervised Learning of Optimal Transport plan prediction (ULOT). We propose a novel architecture based on graph neural networks (GNN) and cross attention mechanisms to predict the OT plan with a complexity of $O(n_1 n_2)$, which is significantly faster than classical solvers. In addition the neural network is conditioned by the parameters of the FUGW problem, which allows to efficiently predict OT plan for all the possible values of the parameters. This is particularly useful in the unbalanced case where the solution is very sensitive to the parameters and validation is often necessary. We show in our experiments, on simulated and real life data, that our method outperforms classical solvers in terms of speed by two orders of magnitude while providing OT plans with competitive loss. The predicted transport plan can also be used as a warm start for classical solvers, which reduces the number of iterations and the overall time of the algorithm. We also show that our method provides a smooth estimation of the transport plan that can be used for numerous applications such as parameter validation for label propagation or gradient descent of a FUGW loss. Our code is available at `https://github.com/smazelet/ULOT`

## 2   Learning to predict OT plans between graphs

In this section we first introduce the FUGW optimal transport problem and its associated loss function. Next we present our amortized optimization strategy called Unbalanced Learning of Optimal Transport (ULOT) and detail the architecture of the neural network. Finally we discuss the related works in deep learning for optimal transport and graph matching.

### 2.1   Fused Unbalanced Gromov Wasserstein (FUGW)

**Definition of the FUGW loss**   Consider the two graphs $\{G_k = (\boldsymbol{F}_k, \boldsymbol{D}_k, \boldsymbol{\omega}_k)\}_{k=\{1,2\}}$ with $n_1$ and $n_2$ nodes respectively. For $k \in \{1, 2\}$, they are characterized by their node features $\boldsymbol{F}_k \in \mathbb{R}^{n_k \times d}$, their connectivity matrices $\boldsymbol{D}_k \in \mathbb{R}^{n_k \times n_k}$ (usually adjacency matrix or shortest path distance matrix) and their node weights $\boldsymbol{\omega}_k \in \Delta_{n_k} \triangleq \{(\omega_k^1, ..., \omega_k^{n_k}), \sum_{i=1}^{n_k} \omega_k^i = 1\}$ that characterize the node's relative importance [34]. Note that in the following we will assume that these weights are uniform, i.e. $\omega_k^i = 1/n_k$ for $i = 1, ..., n_k$. The goal of FUGW [30] is to learn a positive transport plan $\boldsymbol{P} \in \mathbb{R}^{n_1, n_2}$

between the nodes of $G_1$ and $G_2$ that minimizes the following loss function:

$$\mathrm{L}^{\alpha,\rho}(G_1, G_2, \boldsymbol{P}) = (1 - \alpha) \sum_{i,j=1}^{n_1,n_2} \left\| (\boldsymbol{F}_1)_i - (\boldsymbol{F}_2)_j \right\|_2^2 \boldsymbol{P}_{i,j} \tag{1}$$

$$+ \alpha \sum_{i,j,k,l=1}^{n_1,n_2,n_1,n_2} | (\boldsymbol{D}_1)_{i,k} - (\boldsymbol{D}_2)_{j,l} |^2 \boldsymbol{P}_{i,j} \boldsymbol{P}_{k,l} \tag{2}$$

$$+ \rho \left( \mathrm{KL}(\boldsymbol{P}_{\#1} \otimes \boldsymbol{P}_{\#1} | \boldsymbol{\omega}_1 \otimes \boldsymbol{\omega}_1) + \mathrm{KL}(\boldsymbol{P}_{\#2} \otimes \boldsymbol{P}_{\#2} | \boldsymbol{\omega}_2 \otimes \boldsymbol{\omega}_2) \right). \tag{3}$$

The FUGW loss is a combination of the Wasserstein distance (1) that measures the preservation of node features, the Gromov Wasserstein distance (2) that measures the conservation of local geometries and a penalization of the violation of the marginals for the OT plan with the KL divergence (3). The terms (1) and (2) are weighed by the trade-off parameters $\alpha \in [0, 1]$ and the marginal penalization (3) is weighted by $\rho$. Unbalanced OT is a very general and robust framework that can adapt to differences in the geometry of the graph vertices.

**Complexity of solving** $\min_{\boldsymbol{P} \geq 0} \mathrm{L}^{\alpha,\rho}(G_1, G_2, \boldsymbol{P})$   In order to solve the FUGW problem one needs to minimize the FUGW loss with respect to the optimal transport plan $\boldsymbol{P}$. Because of the Gromov Wasserstein term, the complexity of the FUGW loss or its gradient for a given plan $\boldsymbol{P}$ is theoretically quartic $O(n_1^2 n_2^2)$. In the case of the square loss, Peyré et al [21] showed that the complexity can be reduced to cubic complexity $O(n_1 n_2^2 + n_1^2 n_2)$, which remains computationally intensive for large graphs with typically more than $10k$ nodes.

Existing methods to minimize the FUGW loss use a block coordinate descent scheme on a lower bound of the objective that consists in solving at each iteration a linearization of the quadratic problem [8], [30] and requires at each iteration to compute the cubic gradient. In the following we will compare ULOT to three different types of inner-solvers for the linearized problem: the Majorization-minimization (MM) algorithm [6], the inexact Bregman Proximal Point (IBPP) algorithm [39] and a more classical L-BFGS-B algorithm [4]. We will also compare our approach to the entropic regularization of the FUGW loss for which the inner problem can be solved using the Sinkhorn algorithm. All those methods are iterative and require a $O(n_1 n_2^2 + n_1^2 n_2)$ gradient/linearization computation at each iteration. This is a major bottleneck for the FUGW problem, especially when the graphs are large or FUGW has to be solved multiple times, for instance when computing a FUGW barycenter while selecting the parameters $(\alpha, \rho)$ of the method.

## 2.2   ULOT optimization problem and architecture

**Learning to predict FUGW OT plans**   We propose to train a model $\boldsymbol{P}_\theta^{\rho,\alpha}(G_1, G_2)$, parametrized by $\theta$, that, given two graphs $G_1, G_2$ and $(\rho, \alpha)$ parameters can predict a FUGW transport plan, or at least a good solution to the FUGW problem, between them. We want to avoid training the model in a supervised way where for each pair of graphs $(G_1, G_2)$ in the training set we need to pre-compute the corresponding transport plan $\boldsymbol{P}$. This is why we propose to train the model in an unsupervised way using amortized optimization [3]. We do this by sampling pairs of graphs $(G_1, G_2)$ from a training dataset $\mathcal{D}$ and parameters $(\rho, \alpha)$ and minimizing the expected FUGW loss. The ULOT model is trained to minimize:

$$\min_\theta \quad \mathbb{E}_{G_1, G_2 \sim \mathcal{D}^2, \alpha, \rho \sim \mathcal{P}} \left[ \mathrm{L}^{\alpha,\rho}(G_1, G_2, \boldsymbol{P}_\theta^{\rho,\alpha}(G_1, G_2)) \right]. \tag{4}$$

Where $\mathcal{P}$ is the distribution of the parameters $(\rho, \alpha)$ chosen in the experiment as $\mathcal{P}_\rho, \mathcal{P}_\alpha$, where $\mathcal{P}_\rho$ is the log uniform distribution between $10^{-7}$ and 1, and $\mathcal{P}_\alpha$ follows the Beta distribution with parameters $(0.5, 0.5)$. The fact that we optimize the loss over some intervals of the parameters $(\rho, \alpha)$ allows the model to generalize to different values of the parameters to explore or even optimize them.

**Encoding the parameters** $(\rho, \alpha)$   Since we want the model $\boldsymbol{P}_\theta^{\rho,\alpha}(G_1, G_2)$ to depend on the parameters $(\rho, \alpha)$, we need to encode them in a way that can be used by the neural network. To do that, we add them to the node features of the graphs at each layer of the network. $\rho$ is a positive scalar that impacts the mass of the OT plan and can be included as such. But $\alpha$ is a scalar in $[0, 1]$ fixing the tradeoff between the Wasserstein and Gromov-Wasserstein terms that can have a large impact close to 0 and 1. In order to facilitate its use in the neural network, we propose to encode it in a more

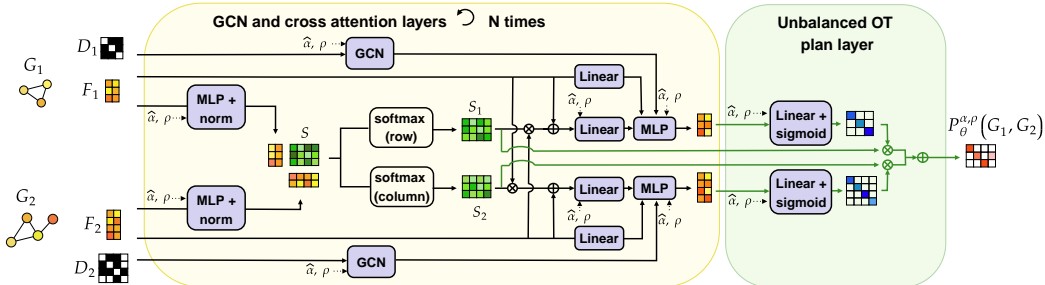

Figure 1: **ULOT architecture for OT plan prediction** The architecture consists of two parts. a node embedding layer repeated $N$ times that relies on cross graph attention and self node updates (GCN), and the final transport plan prediction layer that predicts node weights and the output transport plan.

expressive way with a positional encoding technique. We chose to use the same technique as in [32] using the Fourier basis for encoding time in flow matching applications as follows:

$$\hat{\alpha} = \left[ (\cos(k\pi\alpha))_{k=1,\ldots,d} | (\sin k\pi(1-\alpha))_{k=1,\ldots,d} \right]. \tag{5}$$

where we set $d = 10$ in our experiments. $\hat{\alpha}$ and $\rho$ are concatenated to the node features at each layer of the network as detailed next.

### 2.3 Proposed cross-attention neural architecture

We now detail the proposed neural network architecture that takes as input two graphs $G_1$ and $G_2$ and the parameters $(\rho, \alpha)$ and predicts the transport plan $\boldsymbol{P}_\theta^{\rho,\alpha}(G_1, G_2)$. The proposed ULOT architecture summarized in Figure 1 consists of two main parts. The first part contains $N$ layers of node embedding and attention-based cross-graph interactions that incorporate the geometry of each graph. The final layer predicts the transport plan from the learned node features and interactions with node reweighting. The design choices are further motivated in the ablation studies in Section C.

**Node embedding with cross attention**   The first block is an adaptation of the Graph Matching Network from [13] that outputs node features that are relevant in a matching context. Intuitively the cross attention mechanism computes node features that are similar for nodes that can be matched and dissimilar for nodes that should not. The node embeddings are computed at each layer using two paths. One path is called the *self* path that simply consists of a GCN treating each graph independently. The use of GCNs is motivated by the Gromov term in the FUGW loss that requires a model taking into account the graph topology. For each graph $G_k$ for $k \in \{1, 2\}$ , *self* node features are computed with

$$\boldsymbol{F}_k^{\text{self}} = \text{GCN}(\boldsymbol{F}_k). \tag{6}$$

The second path is the *cross* path that computes a similarity matrix between the node features of both graphs and learns new features that characterize their interactions. In parallel, *cross* node features $\boldsymbol{F}_{1\to 2}^{\text{cross}}$, $\boldsymbol{F}_{2\to 1}^{\text{cross}}$ are computed with an attention block, for $k, k' \in \{1, 2\}, k \neq k'$, learned in the following way:

1. Compute embeddings:
$$\boldsymbol{F}_k^{\text{cross}} = \text{MLP}(\boldsymbol{F}_k, \rho, \hat{\boldsymbol{\alpha}}). \tag{7}$$

2. Compute similarity matrix $\boldsymbol{S}$ and the row/column attention matrices $\boldsymbol{S}_1, \boldsymbol{S}_2$ with, for $s$ the cosine similarity and $i \in [1, n_1], j \in [1, n_2]$:

$$S_{i,j} = s\left((\boldsymbol{F}_1^{\text{cross}})_i, (\boldsymbol{F}_2^{\text{cross}})_j\right) \quad \text{and} \quad \begin{cases} \boldsymbol{S}_1 = \text{softmax}_{\text{row}}(a^2 \boldsymbol{S}) \\ \boldsymbol{S}_2 = \text{softmax}_{\text{column}}(a^2 \boldsymbol{S}) \end{cases} \tag{8}$$

where $a \in \mathbb{R}$ is an hyperparameter.

3. Compute the updated node features $\boldsymbol{F}_{1\to 2}^{\text{cross}}$ and $\boldsymbol{F}_{2\to 1}^{\text{cross}}$:

$$\boldsymbol{F}_{1\to 2}^{\text{cross}} = \text{Linear}\left(\boldsymbol{F}_2^{\text{cross}} - \boldsymbol{S}_2^T \boldsymbol{F}_1^{\text{cross}}\right), \quad \boldsymbol{F}_{2\to 1}^{\text{cross}} = \text{Linear}\left(\boldsymbol{F}_1^{\text{cross}} - \boldsymbol{S}_1 \boldsymbol{F}_2^{\text{cross}}\right), \tag{9}$$

At the end of the layer, for $k, k' \in 1, 2$, the learned node features from the self (GCN) and cross-attention paths are merged with the input node features with the following equation:

$$\boldsymbol{F}_k^{\text{final}} = \text{MLP}\left(\text{Linear}(\boldsymbol{F}_k), \text{Linear}(\boldsymbol{F}_k^{\text{self}}), \text{Linear}(\boldsymbol{F}_{k' \to k}^{\text{match}}), \rho, \hat{\boldsymbol{\alpha}}\right), \tag{10}$$

where the inputs to the MLP are concatenated. The use of those two paths allows to learn node features that take into account both the graph geometry and the cross-interactions between the graphs.

**Transport plan prediction with cross attention and node scaling**    The optimal transport plan prediction block is designed so as to predict an unbalanced OT plan that must contain pairwise relationships between nodes but also that can discard (reweight) nodes that cost too much from an OT perspective. This is done by separating the two aspects. We chose to use for the pairwise relationship in the OT plan the average of the cross attention matrices corresponding to row-wise softmax and a column-wise softmax respectively. This matrix gives a good starting point for the plan, which is then refined in the unbalanced optimal transport plan block, where the scaling per node makes it no longer balanced. This is why we also provide in the last block two node reweighting heads that allow scaling on the OT plan and enable the network to discard mass. These heads predict weight vectors $\boldsymbol{v}_k \in \mathbb{R}^{n_k}, k \in \{1, 2\}$ on the nodes of the graphs that represent how much weight is transported from each node with

$$\boldsymbol{v}_1 = \text{sigmoid}(\text{Linear}(\boldsymbol{F}_1^{\text{final}}, \rho, \hat{\boldsymbol{\alpha}})), \quad \boldsymbol{v}_2 = \text{sigmoid}(\text{Linear}(\boldsymbol{F}_2^{\text{final}}, \rho, \hat{\boldsymbol{\alpha}})) \tag{11}$$

We learn the transport plan from the similarity matrices $S_1, S_2$ of the last layer

$$\boldsymbol{P}_\theta^{\rho, \alpha}(G_1, G_2) = \frac{1}{2}\left(\frac{1}{n_1}\boldsymbol{S}_1 \text{diag}(\boldsymbol{v}_1) + \frac{1}{n_2}\text{diag}(\boldsymbol{v}_2)\boldsymbol{S}_2\right). \tag{12}$$

This allows us to learn a transport plan that is unbalanced and separates the problem of finding the node interactions across graphs, done with the cross attention, and the problem of learning the individual node weights that is specific to unbalanced OT.

## 2.4   Related works

**Deep optimal learning for transport**    Several methods have been proposed for accelerating the resolution of optimal transport problems with deep learning. For instance [26] proposed to model the dual potentials of the entropic regularized OT problem with a neural network. Neural OT [11] learns the classical OT mapping between two distributions using a neural network. Recent Neural OT extensions have also been proposed to solve the GW problem in [19, 38]. This usually allows for solving large OT problems but the resulting neural network is a solution for a specific pair of distributions and needs to be optimized again for new distribution pairs.

Meta OT [2] uses a strategy called amortized optimization [3] to learn an MLP neural network that predicts on the samples the dual potentials of the entropy regularized OT problem. Their model can be used to predict the entropic transport plan between two new distributions using the primal-dual relationship. However, Meta OT cannot be used for Quadratic OT problems such as the Gromov-Wasserstein or FUGW problems because the optimization problem is not convex and the primal-dual relationship is much more complicated [42]. In fact, in order to use an approach similar to Meta OT, one would need to use a linearization of the problem that is $O(n^3)$ which would cancel part of the advantage of using an efficient neural network.

ULOT has been designed to perform OT plan prediction with $O(n^2)$ complexity. Our approach also focuses on graph data. There is a need to go beyond MLP in order to design a neural network that can use the graph structure. In this sense, ULOT can be seen as a generalization of Meta OT to the case of unbalanced OT between graphs. Finally we learn a model that can predict the transport plan conditioned on the parameters of the OT problem, in our case the $(\alpha, \rho)$, which is particularly novel and has not been done before to the best of our knowledge.

**Deep learning for graph matching**    Various neural architectures have been proposed for deep graph matching, which refers to the problem of finding structural correspondence between graphs. However these methods often face notable limitations. Some approaches are limited to predicting a global similarity score between pairs of graphs without predicting node correspondence [13, 14]. Other methods that do provide node-level matching typically rely on supervised training [36, 25, 41], which limits their applications to domains with available ground truth correspondences such as images.

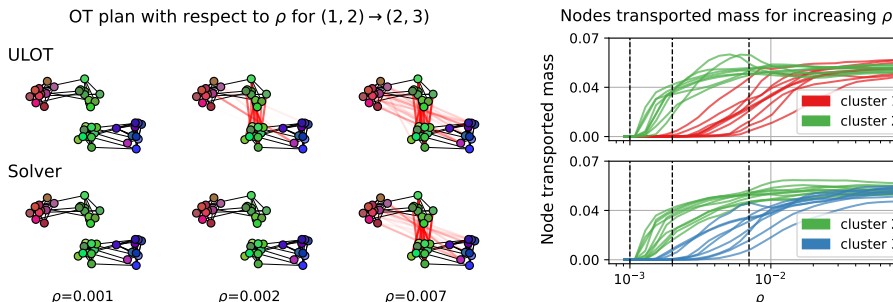

Figure 3: Illustration of OT plans for different values of $\rho$. (left) OT plan (red lines) predicted by ULOT (top) and estimated by the IBPP solver (bottom). The red lines opacity is proportional to the amount of transported mass. (right) Marginals on the nodes of both graphs for different values of $\rho$ colored by the cluster they belong to.

Unsupervised methods, on the other hand, rely on task-specific objectives such as enforcing cycle consistency [33, 37] or adopt contrastive learning approaches by matching graphs to their augmented copies, for which they know the ground truth [15].

To improve the robustness of the matching between graphs of different sizes, some methods discard a subset of correspondences. Various methods have been proposed such as selecting the top-k most confident matches [35] or adding dummy nodes [10]. However, while these methods have a similar goal to ULOT they perform a hard selection of nodes, which is useful in the case of outliers but not when we need a more continuous way to reduce some node importance. Finally, while our method bears some similarities with neural networks for graph matching, it is important to note that the objective of Unbalanced OT is fundamentally different from graph matching.

## 3 Numerical experiments

In this section, we evaluate ULOT and compare it to classical solvers on both a simulated dataset of Stochastic Block Models (SBMs) with different numbers of clusters and on the Individual Brain Charting (IBC) dataset [23] of functional MRI activations on brain surfaces.

### 3.1 Illustration and interpretation on simulated graphs

**Dataset and training setup** We first train ULOT on a simulated dataset of Stochastic Block Models (SBMs) with 3 linearly connected clusters and 3D node features that are a one hot encoding of the cluster classes $1, 2$ or $3$ with additive centered Gaussian noise. To investigate the properties of the transport plans learned by ULOT in comparison to those estimated by classical solvers, we construct three types of graphs with different clusters. The first type includes graphs where all three clusters $(1, 2, 3)$ are present, the second type of graphs has clusters $(1, 2)$ and the third has clusters $(2, 3)$. All graphs in the dataset have a random number of nodes ranging from $30$ to $60$ and the training dataset consists of $50000$ simulated pairs $(G_1, G_2, \rho, \alpha)$.

With those three types of graphs, unbalanced OT should be able to find a transport plan that

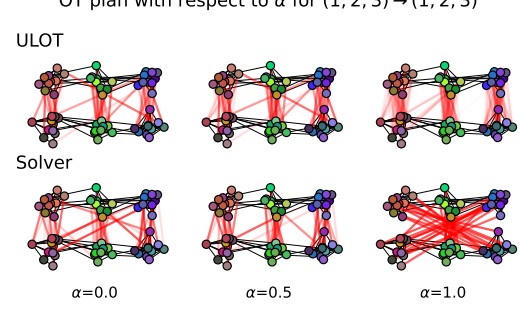

Figure 2: Examples of transport plans (red lines) predicted by ULOT (top) and estimated by the IBPP solver (bottom) for different $\alpha$ values. The red lines opacity is proportional to the amount of transported mass.

matches the clusters if they are present in both graphs, but discard clusters that are not shared (when $\alpha, \rho$ are properly selected).We compare ULOT to the numerical solver IBPP, which provides a good trade off between performance and speed.

**Regularization path with respect to the parameters $\rho$ and $\alpha$**   We first illustrate the effect of $\rho$ on the predicted transport plan between graphs containing clusters $(1, 2)$ and $(2, 3)$. We see in Figure 3 (left) that the transported mass increases with $\rho$ for both ULOT and the solver but while the solver is stuck in a $0$ mass local minimum, ULOT is able to find a plan with mass and lower cost for $\rho = 0.002$. We also provide in Figure 3 (right) the regularization path for the marginals on the nodes of both graphs which shows that the nodes from cluster 2 in green are the first to receive mass when $\rho$ increases finding a proper alignment of the clusters.

Next we illustrate the OT plans between two graphs of type $(1, 2, 3)$ for different values of $\alpha$ in Figure 2. Because the node features are noisy, both the Gromov Wasserstein and Wasserstein terms in the loss are needed to predict accurate plans. We see that while the transport plans from ULOT and the solver are comparable for $\alpha < 1$, the solver wrongly matches the clusters for $\alpha = 1$ because it does not use the node feature information and can permute classes. ULOT does not permute the clusters for $\alpha = 1$ because it is continuous w.r.t. $\alpha$ and has learned to use the node features to find a better plan.

**Optimizing the hyperparameter for a prediction task**   We now consider the task of label propagation between graphs, where node labels are known on a source graph but partly missing on a target graph. To infer the missing labels, we transport the one hot encoding of the source graph node labels onto the target graph, producing label probabilities for each node similarly to what was proposed in [28, 24]. A key challenge of this approach, when the graph types can differ, lies in selecting the appropriate FUGW parameters $(\rho, \alpha)$ to ensure that the plan is relevant for label propagation.

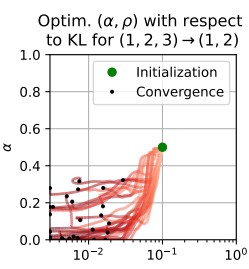

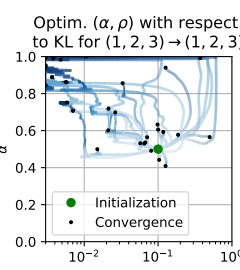

Figure 4: $(\alpha, \rho)$ optimization trajectories for (top) different types $(1, 2, 3) \rightarrow (1, 2)$, (bottom) same type $(1, 2, 3)$.

Thanks to our efficient ULOT framework, we can easily compute and visualize the accuracy of the label propagation task as a function of the parameters $(\rho, \alpha)$ for different pairs of graphs. Due to lack of space this is provided in the supplementary material in Figure 11. We find that the accuracy surfaces are relatively smooth and that the optimal parameters greatly depend on the types of graphs transported.

Since ULOT OT plans are by construction fully differentiable with respect to $\rho$ and $\alpha$, we propose to optimize them, but taking as objective a classical smooth proxy for the accuracy: the Kullback-Leibler divergence between the one-hot encoded target classes and the predicted label scores. We show the optimization trajectories of parameters $(\rho, \alpha)$ for different simulated pairs of graphs in Figure 4. We see that the trajectories vary significantly between the two types of graph pairs. Indeed between different types, $\rho$ has to be small to avoid mass transfer between clusters that are not present in both graphs, while for the same types, $\rho$ can be larger to allow mass transfer between clusters that are present in both graphs. Interestingly, we see that for the same graph pair types, the general trend is similar but the trajectories converge to different values, underlining the necessity of parameter validation for individual pairs. This proof of concept shows that ULOT parameters can be optimized for a given task, allowing for efficient bi-level optimization when the inner optimization problem is a FUGW.

**Optimizing a graph wrt the FUGW loss**   One very interesting aspect of our method is that the predicted transport plan is fully differentiable with respect to the graph structure and features. We illustrate this by optimizing a functional of a graph. Given a target graph $G^\star$, we optimize the function $F(G) = L^{\alpha,\rho}(G, G^\star, \boldsymbol{P}_\theta^{\rho,\alpha}(G, G^\star))$ where we expect the graph $G$ to converge to or close to the graph $G^\star$. Starting from $G_0 = G$, at each time step $t$, we predict the transport plan $\boldsymbol{P}_\theta^{\rho,\alpha}(G_t, G^\star)$ between $G_t$ and $G^\star$, compute the associated FUGW loss and update the nodes features and shortest path distance matrix of $G_t$ using backpropagation. We recover the adjacency matrix at time step $t + 1$ by thresholding the shortest path distance matrix. The influence of the tradeoff parameters $\alpha$ is shown on Figure 5. The trajectory for $\alpha = 0.5$ shows that the graph converges to a two-cluster graph with proper labels. In contrast, the trajectory for $\alpha = 1$ shows that the graph converges to a two-cluster

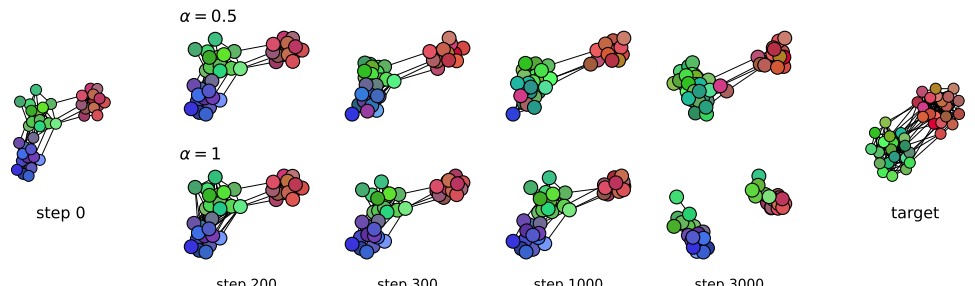

Figure 5: Gradient descent steps of the minimization of of the ULOT FUGW loss between a source graph (step 0) and a target graph for $\alpha = 0.5$ (top) and $\alpha = 1$ (bottom).

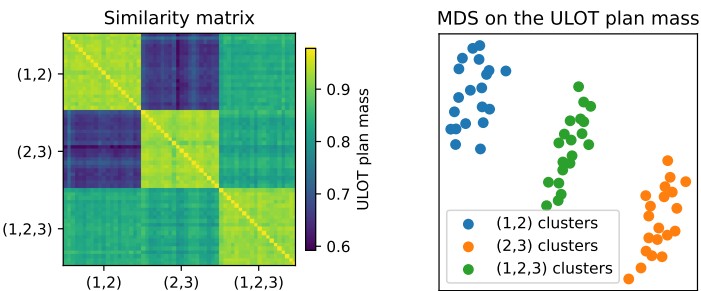

Figure 6: (left) Similarity matrix of the ULOT transport mass between simulated SBM graphs with respective clusters $(1, 2)$, $(2, 3)$ and $(1, 2, 3)$ sorted by type. (left) MDS of the similarity matrix.

graph with wrong labels. This occurs because only the Gromov-Wasserstein term is optimized, so the node features are not updated.

**Using the ULOT transport plan mass**  The FUGW loss computed with the ULOT plan provides a meaningful distance between graphs, but its computation is $O(n_1^2 n_2 + n_2^2 n_1)$, where $n_1, n_2$ are the number of graph nodes. In contrast, computing the ULOT transport plan only has a quadratic time complexity. In unbalanced OT, the total mass of the OT plan $m(\boldsymbol{P}) = \sum_{i,j} P_{i,j}$ will decrease when the two graphs are very different, since in this case the marginal violation will cost less than the transport cost. This is why we propose to use the ULOT transport plan mass as a graph similarity measure (positive and between 0 and 1).

We evaluate this approach on synthetic SBM drawn from three cluster configurations: $(1, 2)$, $(2, 3)$ and $(1, 2, 3)$. We compute the ULOT transport plan using fixed hyperparameters $\alpha = 0.5$ and $\rho = 0.01$. As shown in Figure 6 (left), the resulting similarity matrix reveals the structure of the dataset. This similarity matrix can naturally be used for (spectral) graph clustering, or even for dimensionality reduction and visualization as illustrated in Figure 6 (right) with multidimensional scaling of the similarity matrix where the relation between the types of graphs is clearly recovered.

## 3.2  Solving FUGW for Functional MRI brains

**Dataset description**  We now evaluate ULOT on the Individual Brain Charting (IBC) dataset [23] which is a dataset of functional MRI activations on brain surfaces. The dataset is made of surface meshes with $160k$ vertices associated with different fMRI activations. As the dataset focuses on individual information, it includes subject-specific brain geometrical models, associated with individual fMRI activations. This justifies the necessity of the unbalanced framework to adapt mass between regions that can vary in size between subjects [30].

**Experimental setup**  The high dimensionality of the meshes in the dataset makes it particularly interesting for data augmentation. In order to obtain more training graphs, we perform a parcellation using Ward algorithm [29], and reduce the graph size to 1000 nodes, where the node features are the fMRI activations averaged over the grouped vertices. Data augmentation consists of generating 10 different graphs for each of the 12 subjects, as the results of the Ward clustering using randomly

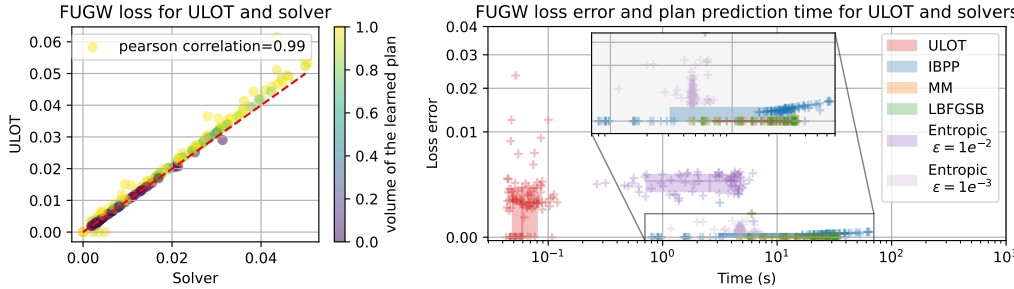

Figure 7: (left) Comparison of the loss obtained with ULOT and IBPP solver, the dashed lines correspond to equality. (right) Plot of loss error VS time for ULOT and other solvers. Colored squares correspond to the 20-80% quantiles for both measures.

sampled activations. The geometric information matrix $D$ is the shortest path distance matrix and the 3D node positions are concatenated with the node activation features to provide a positional encoding. We construct a dataset of the $14400$ different graph pairs and train the network using a $60/20/20$ train/val/test split. We provide more details on the experimental setup in Section E.

**Comparison to solvers in terms of loss and computational time**    We first compare the FUGW loss of OT plans predicted by ULOT and the IBPP solver. We find in Figure 7 (left) that both losses are very close and highly correlated with a Pearson correlation of $0.99$.
Next we compare the loss error (wrt the best among all solvers) of ULOT and the other solvers introduced in section 2.1, namely IBPP [39], MM [6], LBFGSB and Sinkhorn for the entropic regularized FUGW [8, 30] using the Python library POT [9]. We find in figure 7 (right) that even though ULOT makes errors, it is up to $100$ times faster than classical solvers and $10$ times faster than Sinkhorn for a smaller error.
This computational gain on graphs of size $1000$ is very important as the solvers have a cubic time complexity with respect to the number of nodes, while ULOT has a quadratic time complexity as shown in Figure 8 (left) where computation time is plotted against the number of nodes.

**ULOT as warmstart.** Finally when high precision is needed, we can use ULOT as a very efficient warm start for the IBPP solver. We find in Figure 8 (right) that using ULOT as a warm start allows the solver to converge much faster. This means that if high precision is required, using ULOT as a warmstart for a solver is an efficient alternative.

**fMRI activations prediction using ULOT plans**    We now illustrate the use of ULOT transport plans on fMRI data for activation prediction between brain graphs. We use the same experimental setup as in [30] where fMRI activations from a source brain graph $G_1$ are transported to predict activations on a target brain graph $G_2$. First, we compute the ULOT transport plan $P_\theta^{\alpha,\rho}$ between the two graphs using the model trained above. Then, we predict for a new mental task, *colorless_auditory* contrast from MathLanguage, the fMRI activations $\widehat{F_2}$ on the nodes of $G_2$ by transporting the activations $F_1$ from graph $G_1$:

$$\widehat{F_2} = \text{diag}\left(\frac{1}{(P_\theta^{\alpha,\rho})_{\#2}}\right)(P_\theta^{\alpha,\rho})^\top F_1. \tag{13}$$

We visualize in Figure 9 the activations on the parcelled brain regions and observe that the general fMRI trend is conserved through the transportation. While this is only a qualitative experiment, this opens doors for future use of OT in large scale experiments on fMRI data.

## 4   Conclusion, limits and future work

We have introduced ULOT, a new unsupervised deep learning approach for predicting optimal transport plans between graphs, trained by minimizing the FUGW loss. We have shown that ULOT is able to predict transport plans with low error on both simulated and real datasets, up to $100$ times faster than classical solvers. Its low complexity and differentiability make it naturally efficient for

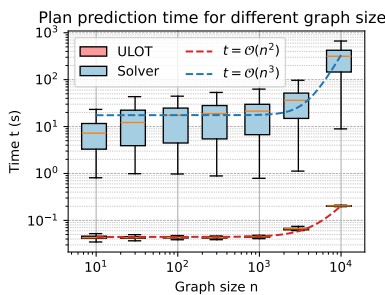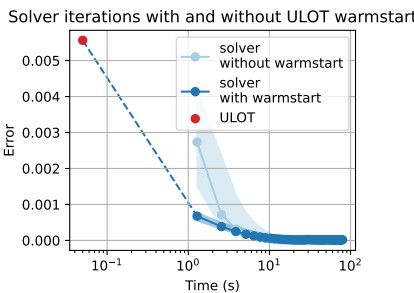

Figure 8: (left) FUGW transport plan prediction time for ULOT and IBPP solver for different graph sizes. (right) IBPP solver loss along iterations with and without ULOT warmstart, reported with the $20\% - 80\%$ quantiles.

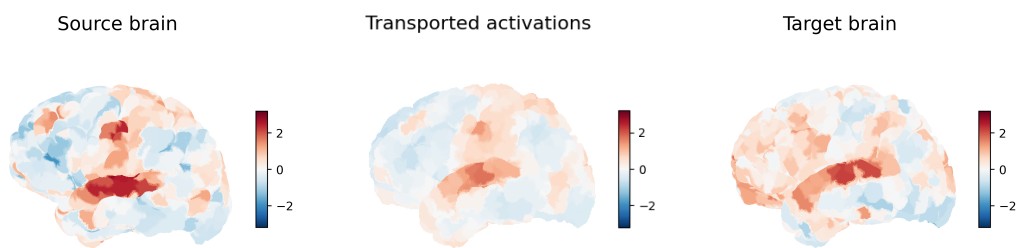

Figure 9: Example of transporting *colorless_auditory* activations from the MathLanguage contrast between two subjects from the IBC dataset. Each brain is equipped with its individual geometry
.

minimizing functionals of optimal transport plans and performing FUGW parameter selection. ULOT also allows for discovering novel ways to use FUGW OT plans such as using their total mass as a measure of similarity of complexity $O(n^2)$ between graphs of different sizes.

While we believe that ULOT is a very promising step towards the use of deep learning for optimal transport, we also acknowledge its limitations and propose research directions for addressing them. The very fast prediction comes at the cost of a small error in the predicted transport plans, which can limit its applications in a context where high precision is needed. While this can be avoided by using ULOT as a warmstart for a solver, there is still room for improvement for directly predicting even more accurate plans. Also we were limited in our experiments to graphs of size $n \leq 10000$ due to GPU memory constraints and going further might require dedicated developments such as lazy tensors or other memory-efficient techniques [7] for cross-attention. In the future, we plan to apply this method to large-scale applications such as activation prediction on high-resolution brain surfaces and computation of graph barycenters. While these applications require large training datasets, which is not the norm for fMRI data, we plan to further investigate our random parcellation data augmentation technique to train a more general model that can be effective across subjects.

## Acknowledgments and Disclosure of Funding

This work was granted access to the HPC resources of IDRIS under the allocation 2025-AD011016350 made by GENCI. This work is supported by Hi! PARIS and ANR/France 2030 program (ANR-23-IACL-0005). This research was also supported in part by the French National Research Agency (ANR) through the MATTER project (ANR-23-ERCC-0006-01). This work benefited from state aid managed by the Agence Nationale de la Recherche under the France 2030 programme, reference ANR-22-PESN-0012 and from the European Union's Horizon 2020 Framework Programme for Research and Innovation under the Specific Grant Agreement HORIZON-INFRA-2022-SERV-B-01, grant agreement number 10.3030/101147319. Finally, it received funding from the Fondation de l'École polytechnique. We thank Quang Huy Tran for providing code for the FUGW solvers and for helpful discussions.

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

Figure 10: (left) Comparison of the loss obtained with ULOT and IBPP solver for the simulated graphs, the dashed line corresponds to the equality. (right) Transport plan values for increasing values of $\rho$, colored by whether they link common or different clusters.

## A   Training setup details

**Compute resources**   We trained our network on an NVIDIA V100 GPU for $100$ hours on the IBC dataset and a few hours on the smaller simulated dataset. Note that while the network is $O(n^2)$ with $n$ the number of graph nodes, the main training bottleneck comes from the need to compute the $O(n^3)$ FUGW loss for each transport plan at every epoch.

**Hyperparameters**   The hyperparameters used for training ULOT on both the simulated graphs and the IBC dataset are reported in Table 1. All the MLP and GMN have one hidden layer, with weights shared between the two graph branches. Moreover, the first MLP in the node embedding layer preserves the dimensionality of the input node features. Hyperparameter optimization was performed on a subset of the training data using the Optuna library [1]. The code is available in the supplementary materials and will be released on github upon publication. We will also share the pre-trained model weights for both datasets.

Table 1: ULOT hyperparameters

| Hyperparameter | Simulated dataset | IBC dataset |
|---|---|---|
| Learning rate | 0.001 | 0.0001 |
| Batch size | 256 | 64 |
| Optimizer | Adam | Adam |
| Number of node embedding layers $N$ | 5 | 3 |
| Embedding dimension for $\alpha$ | 10 | 10 |
| Node embedding layer final out dimension | 256 | 256 |
| MLP hidden dimension | 64 | 256 |
| GCN hidden dimension | 16 | 128 |
| Temperature value $a$ | 3 | 5 |

## B   Additional experiments on the simulated graphs

**Comparison of FUGW loss for ULOT and solver on the simulated graphs**   We train ULOT on the dataset of simulated SBMs introduced in Section 3.1 and test it on new pairs of simulated graphs sampled from the same distribution. We find in Figure 10 (left) that similarly to the IBC dataset ULOT finds transport plans that have a FUGW loss perfectly correlated with the FUGW loss obtained with the IBPP solver.

**Regularization path of the transport plan component $P_{i,j}$ with respect to $\rho$**   We predict ULOT transport plans between a pair of graphs with cluster configurations $(1, 2)$ and $(2, 3)$ for different values of $\rho$ and show the regularisation path of each transport plan entry $(P_\theta^{\alpha,\rho})_{i,j}$ for $i \in [1, n_1]$ and $j \in [1, n_2]$ in Figure 10 (right). We observe that transport plan values corresponding to nodes in the shared cluster 2 increase more rapidly with $\rho$ compared to values between nodes in non-overlapping clusters. Moreover, we find that there exists an optimal value around $\rho \simeq 0.05$, for which transport

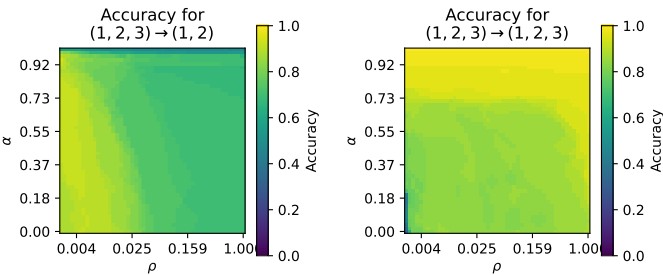

Figure 11: Label propagation accuracy of the ULOT FUGW transport w.r.t. $(\rho, \alpha)$ between (left) different types $(1, 2, 3) \rightarrow (1, 2)$, (right) same type $(1, 2, 3)$.

plan mass is predominantly assigned to entries corresponding to the common cluster, effectively discarding irrelevant correspondences.

**Visualization of the accuracy for label propagation**   We visualize the accuracy surfaces on Figure 11 for the label propagation task introduced in Section 3.1 for a range of $(\alpha, \rho)$ values. We consider two different types of pairs: pairs with clusters $(1, 2, 3)$ and $(1, 2)$ and pairs with clusters $(1, 2, 3)$. We see that the accuracy surfaces are smooth and that the optimal parameter values differ across pair types. We optimize $\rho$ and $\alpha$ using gradient descent on each pair of graphs by minimizing the KL divergence between the predicted class probabilities obtained with the ULOT transport plan and the ground truth target one hot encodings of the classes on $50\%$ of the nodes. We obtain an accuracy of $0.87 \pm 0.092$ on pairs with similar clusters $(1, 2, 3)$ and $0.74 \pm 0.11$ on the pairs with different clusters.

## C   Ablation studies

We provide an ablation study of the impact of each step of the OT plan block on the SBM dataset in Table 2. As a first ablation we remove the node scaling in the OT plan head, as a second ablation we replace the whole OT plan head by a much simpler nonlinear transformation of the node embedings to recover an OT plan with $(P_\theta^{(\alpha, \rho)})_{i,j} = \frac{1}{1 + ||(F_1^{\text{final}})_i - (F_2^{\text{final}})_j||_2^2}$, where $F_1^{\text{final}}$ and $F_2^{\text{final}}$ are the outputs of the GCN and cross attention block for the first and second graphs respectively.

Table 2: Ablation studies

| Ablation | Relative error | Pearson correlation |
|---|---|---|
| None | $0.19 \pm 0.29$ | 1.0 |
| Remove node scaling | $4.1 \pm 9.5$ | 0.89 |
| Replace OT plan block by nonlinearity | $27 \pm 49$ | 0.48 |

We can see that node scaling is essential for recovering unbalanced OT plans especially for small $\rho$ parameters (unbalanced plans that do not sum to one), and removing them increases the error (but remains surprisingly well correlated). Using a simpler output head for the OT plan leads to a failure to predict OT plans.

## D   Robustness to distribution shifts

We tested the following distribution shift: we trained ULOT on a dataset of Stochastic Block models with balanced clusters (every cluster has the same number of nodes), and tested the model on a dataset of SBMs with unbalanced clusters, where the cluster ratio is sampled following a Dirichlet distribution. We report the results Table 3 and find that even though the relative loss error increases, the Pearson correlation between the ground truth losses and the ULOT losses remains stable.

Table 3: Distribution shifts

| Distribution | Relative error | Pearson correlation |
|---|---|---|
| Balanced clusters (train distribution) | $0.19 \pm 0.29$ | 1.0 |
| Unbalanced clusters (distribution shift) | $0.27 \pm 0.69$ | 0.98 |

# E   Illustration of the fMRI alignement

**Experimental details on the IBC dataset**   We use brain cortical surfaces from the IBC dataset consisting of approximately 160k vertices, each associated with fMRI contrasts obtained as subjects perform specific tasks. To train ULOT, we use the 239 contrasts from the tasks in Table 4. The predicted activation visualized on Figure 9 is selected from the 30 left out contrasts from the MathLanguage task. For each subject, we perform a parcellation of the surface to form 1000 brain regions, using the Ward's hierarchical clustering algorithm. Each contrast is then averaged over the vertices in every brain region. From this parcellation, we construct 1000 node graphs, where edges connect spatially adjacent regions. Each node has a 242 dimensional feature vector composed of the region's contrasts concatenated to its 3D node position.

To augment the dataset, we generate multiple parcellations for each subject by randomly selecting 20% to 40% of the tasks, which produces variability in the geometries and the activations across all generated brains. The final dataset is constructed from all possible graph pairs, which we randomly split into 60% training, 20% validation, and 20% test sets.

Table 4: IBC tasks used for alignment

| Task |
|---|
| ArchiEmotional |
| ArchiSocial |
| ArchiSpatial |
| ArchiStandard |
| HcpEmotion |
| HcpGambling |
| HcpLanguage |
| HcpMotor |
| HcpRelational |
| HcpSocial |
| HcpWm |
| RSVPLanguage |
| PreferenceFaces |
| PreferenceHouses |
| PreferenceFood |
| PreferencePaintings |
| MCSE |
| Moto |
| Visu |
| Audi |
| MVEB |
| MVIS |
| Lec1 |
| Lec2 |
| TheoryOfMind |
| PainMovie |
| EmotionalPain |
| Enumeration |
| VSTM |
| Self |

