# OpenReview forum: "Unsupervised Learning for Optimal Transport plan prediction between unbalanced graphs"
_NeurIPS.cc/2025/Conference — NeurIPS 2025 poster_

### Official Review · Reviewer_j6iH · 2025-06-12

**Clarity:** 3
**Significance:** 2
**Originality:** 3
**Rating:** 4
**Confidence:** 3

**Summary:**

This paper proposes a new approach for predicting optimal transport plans between graphs, designed to improve the computational complexity (in time) with respect to the number of nodes of the graphs, in comparison to standard solvers.  It proposes to use a GNN with attention mechanism as model, and the FUGW distance as loss function. The method exhibits a computational complexity of $\mathcal{O}(n^2)$ instead of the $\mathcal{O}(n^3)$ obtained by standard solvers when computing the OT plan between two graphs with n nodes. The authors evaluated the methods on both synthetic and real-world problems, confirming computational gains while maintaining comparable FUGW losses.

**Questions:**

1) How do the authors believe their model compares to simpler neural network models, or how does it relate to other deep learning approaches for optimal transport? If such comparisons are not provided, how should one assess the architecture's performance? It might help if the authors included comparisons or discussions in this regard.

2) While the authors demonstrate various features of their model through synthetic experiments, could the authors elaborate on potential real-world applications?

3) The approach of encoding the weights, specifically rho and alpha, as node features utilizing the Fourier basis would benefit from additional clarification. Could the authors offer further justification or elaboration on this methodology?

**Ethical Concerns:**

["NO or VERY MINOR ethics concerns only"]

**Final Justification:**

The paper presents a technically sound approach that is relevant to the field. While additional comparisons could strengthen the empirical evaluation and improve the overall clarity of the contribution, I appreciate the authors' thoughtful and detailed responses, which address my concerns with seriousness and care. In light of this, I am maintaining my positive evaluation.

**Limitations:**

yes

**Paper Formatting Concerns:**

No formatting issues.

**Quality:**

3

**Strengths And Weaknesses:**

This paper introduces a novel approach for predicting FUGW transport plans. The authors present compelling experimental results using a real-world dataset of fMRI activations. Additionally, the method is  thorough examined through a synthetic problem, offering valuable insights into its behavior. When benchmarked against standard solvers, the proposed method achieves results of comparable accuracy but with significantly reduced computational time. The paper is clearly presented. In essence, there is no major weaknesses. The paper effectively combines deep learning techniques with conclusive experimental results. However, while the paper successfully conveys the potential of using neural networks for predicting FUGW OT plans, it does not compare its approach to other neural network architectures, which makes it challenging to assess the relevance and advantages of the proposed architecture.

---

> ### Author Rebuttal · Authors · 2025-07-30
>
> We thank you for your thoughtful review. We appreciate your questions and have done our best to answer them below.
>
>
> > How do the authors believe their model compares to simpler neural network models, or how does it relate to other deep learning approaches for optimal transport? If such comparisons are not provided, how should one assess the architecture's performance? It might help if the authors included comparisons or discussions in this regard.
>
>
>
> To the best of our knowledge, there are no other deep learning approaches for predicting the FUGW optimal transport plan. Existing neural OT methods (except Meta OT [2]) use a neural network that is optimized to solve a unique OT problem but cannot be used on new data without a new optimization. We originally tried other simpler designs, but their limited performances led to the current architecture.
>
> We provide an ablation study of the impact of each step of the OT plan block in the table below. As a first ablation we remove the node scaling in the OT plan head, as a second ablation we replace the whole OT plan head by a much simpler nonlinear transformation of the node embedings to recover an OT plan with  ($P_\theta^{(\alpha, \rho)})_{i,j} = \frac{1}{1+||(F^{\text{final}}_1)_i -(F^{\text{final}}_2)_j||^2_2},$ where $F^{\text{final}}_1$ and $F^{\text{final}}_2$ are the outputs of the GCN and cross attention block for the first and second graphs respectively.
>
>
> | Ablation |   Relative error   |  Pearson correlation |
> | -------- | -------- | -------- |
> |  None   |   $0.19 \pm 0.29$  |  $1.0$ |
> | Remove node scaling | $4.1 \pm 9.5$    |  $0.89$  |
> | Replace OT plan block by nonlinearity | $27 \pm 49$   |  $0.48$  |
>
> We can see that node scaling is essential for recovering unbalanced OT plans especially for small $\rho$ parameters (unbalanced plans that do not sum to one), and removing them increases the error (but remains surprisingly well correlated). Using a simpler output head for the OT plan leads to a failure to predict OT plans.
>
>
> > While the authors demonstrate various features of their model through synthetic experiments, could the authors elaborate on potential real-world applications?
>
> Our model can be applied to any scenario where there is structured data, such as genomics [A] or neuroimaging [30]. In particular, we are interested in real-world applications in neuroimaging where FUGW-based methods have proven to be effective [30] but are still limited by their computational time.
>
> More precisely, we will use in future works ULOT to predict fMRI activations faster, in scenarios where one subject has many recorded fMRI activations and another has few. As shown in [30] we can transport the activations from the first to the latter. Moreover, we can compute the barycenters of populations of subjects in order to get a typical fMRI recording. This is useful in practice but unscalable using solvers because it requires solving many FUGW problems along the optimization process.
>
> Beyond computational speed, ULOT enables new capabilities for neuroimaging applications such as efficient FUGW hyperparameter selection, which are hard to tune in practice and highly influence the performance.
>
>
> > The approach of encoding the weights, specifically rho and alpha, as node features utilizing the Fourier basis would benefit from additional clarification. Could the authors offer further justification or elaboration on this methodology?
>
> We designed the model so that the information about $\rho$ and $\alpha$ is not lost along the layers, which is a common issue when models are conditioned on a real value, such as time in diffusion models. Inspired by the method in [B], which has become a standard practice for diffusion models, we concatenate the value of rho and the encoding for alpha to the node features at the entry of each MLP, GCN and Linear layer. We found that concatenating to the node features is a simple and effective way of keeping the parameters information.  This design ensures that all layers have direct access to the conditioning parameters and can learn themselves at which layer this information is the most pertinent.
>
> Besides, parameters $\alpha$ and $\rho$ have very different influences on the optimal transport plan: $\rho$ influences the amount of mass that can be discarded, while $\alpha$ has a more subtle role by controlling the tradeoff between the Wasserstein and Gromov Wasserstein terms. They also have very different theoretical dynamics with $\rho\in[0,+\infty]$ and $\alpha \in[0,1]$. This is why we encode $\alpha$ in a more complex way, using a method strongly inspired by the standard positional encodings in Transformers [B]. Here, we find that using $(\\cos(k\\pi\\alpha))\_{k=1,\\dots,d}$ and $(\\sin(k\\pi(1-\\alpha))\_{k=1,\\dots,d}$  improves training while capturing key properties. We also encode both $\alpha$ and $1-\alpha$ to ensure that the dynamics are the same close to $0$ and $1$.
>
>
>
> ### References
>
>    [30] Alexis Thual, Quang Huy Tran, Tatiana Zemskova, Nicolas Courty, Rémi Flamary, Stanislas Dehaene, and Bertrand Thirion. Aligning individual brains with fused unbalanced gromov wasserstein. Advances in neural information processing systems, 35:21792–21804, 2022.
>
>    [A] Huizing, G. J., Peyré, G., & Cantini, L. (2022). Optimal transport improves cell–cell similarity inference in single-cell omics data. Bioinformatics, 38(8), 2169-2177.
>
>    [B] Vaswani, A., Shazeer, N., Parmar, N., Uszkoreit, J., Jones, L., Gomez, A. N., ... & Polosukhin, I. (2017). Attention is all you need. Advances in neural information processing systems, 30.
>
>    [C] Ho, J., Jain, A., & Abbeel, P. (2020). Denoising diffusion probabilistic models. Advances in neural information processing systems, 33, 6840-6851.
>
>    [2] Amos, B., Cohen, S., Luise, G., & Redko, I. (2022). Meta optimal transport. arXiv preprint arXiv:2206.05262.

---

> > ### Comment · Reviewer_j6iH · 2025-08-04
> >
> > Thank you for your response and the efforts undertaken. It has addressed my concerns satisfactorily, and I will maintain my positive evaluation.

---

### Official Review · Reviewer_HUXF · 2025-06-12

**Clarity:** 2
**Significance:** 3
**Originality:** 3
**Rating:** 3
**Confidence:** 4

**Summary:**

This paper proposed a deep learning framework to solve the problem of optimal transport between graphs. Instead of using traditional solution methods, the proposed work can use deep neural network to predict the transportation faster than the existing method. They also showed their model's efficiency on both simulated and real-world (fmri) data.

**Questions:**

1. D1, D2 input are the same for different GCN and cross attention layers or they are updated?
2. What is the training time/computational complexity of the model?
3. What is the transferability of the model? For instance, in your simulated experiment, if you train a model with node number in cluster 1, n1, and node number in cluster 2, n2, does the model still work when you change the number in cluster 1 and the number in cluster 2 relatively but keep the total number the same (e.g., n1+2, n2-2)? Or you need to train the model every time you are facing a new environment?
4. \alpha being 0 or 1 are two extreme numbers in the results. What about 0.2 and 0.8? I would like to know whether features or short distance path are the dominant factor in the deep learning model, and how the results different from the ground truth?
5. The similarity matrix overall had a quite high correlation value (> 0.6). Can this be caused by the sparsity? A diagnal-block high value correlation can be easily affected by the zero connection between different clusters. Also not clear about what the reason of showing fig. 6. MDS (fig 6 right) is not defined in the paper.
6. An example of transportation plan of real-world data from ULOT and solver will help verify the model, especially if authors can also correlate the results with the biological reasoning.

**Ethical Concerns:**

["NO or VERY MINOR ethics concerns only"]

**Final Justification:**

The main revision of the paper should focus on the motivation, idea and logics of the proposed model and its structure. In the current version, simplifying the optimal transport calculation is important, but it is also important to let people understand why and how it works. With additional explanation and experiments shown by the authors, I will increase my rating to 4.

**Limitations:**

Yes.

**Paper Formatting Concerns:**

No.

**Quality:**

3

**Strengths And Weaknesses:**

Strength:
Use a deep neural network to approximate the complex FUGW solver.
The computational complexity in the testing phase is much less than the traditional methods.
With this warm-start of the model, the traditional solver can reach much lower-error results.

Weakness:
The explanation of the model is not enough and unclear. The whole section 2.3 can be compressed into an algorithm format, and authors should explain why they had this structure or information flow, e.g., can author explain why the formation of unbalanced OT plan layer gives the transportation plan?
More results should be added to verify the model. Please see questions.

---

> ### Author Rebuttal · Authors · 2025-07-30
>
> We thank you for your thoughtful review and the important questions you raised.
>
> > The explanation of the model is not enough and unclear. The whole section 2.3 can be compressed into an algorithm format, and authors should explain why they had this structure or information flow, e.g., can author explain why the formation of unbalanced OT plan layer gives the transportation plan? More results should be added to verify the model. Please see questions.
>
>
> We appreciate your question, as it is fundamental that the model design be clearly motivated and we will improve this aspect in the final version. We designed the model so that it can learn transport plans that respect the conditions for minimizing the FUGW loss (unbalanceness, feature and structure integration). We originally tried other simpler architectures, but their limited performance led to the current architecture that we detail below.
>
> * The first block outputs node features that are relevant in a matching context, and is an adaptation of [13]. Intuitively, the cross attention mechanism computes node features that are similar for nodes that can be matched and dissimilar for nodes that should not. The use of GCNs is motivated by the Gromov term in the FUGW loss that enforces that the model should take into account the graph topology (features are already used in MLP and cross-attention).
> * The optimal transport plan prediction block is designed so as to predict an unbalanced OT plan that must contain pairwise relationships between nodes but also that can discard (reweight) nodes that cost too much from an OT perspective. This is done by separating the two aspects:
>     *  We chose to use for the pairwise relationship in the OT plan the average of the cross attention matrix to which we apply a row-wise softmax and a column-wise softmax respectively.
>     * The pairwise matrix above gives a good starting point for the plan, which is then refined in the unbalanced optimal transport plan block, where the scaling per node makes it no longer balanced. This is why we also provide in the last block two node reweighting heads (Linear+sigmoid and multiplication of the plans by the diagonal weight matrices) that allow scaling on the OT plan and enable the network to discard mass.
>
> We provide an ablation study of the impact of each step of the OT plan block on the SBM datasets in the table below. As a first ablation we remove the node scaling in the OT plan head, as a second ablation we replace the whole OT plan head by a much simpler nonlinear transformation of the node embedings to recover an OT plan with  ($P_\theta^{(\alpha, \rho)})_{i,j} = \frac{1}{1+||(F^{\text{final}}_1)_i -(F^{\text{final}}_2)_j||^2_2},$ where $F^{\text{final}}_1$ and $F^{\text{final}}_2$ are the outputs of the GCN and cross attention block for the first and second graphs respectively.
>
>
> | Ablation |   Relative error   |  Pearson correlation |
> | -------- | -------- | -------- |
> |  None   |   $0.19 \pm 0.29$  |  $1.0$ |
> | Remove node scaling | $4.1 \pm 9.5$    |  $0.89$  |
> | Replace OT plan block by nonlinearity | $27 \pm 49$   |  $0.48$  |
>
> We can see that node scaling is essential for recovering unbalanced OT plans especially for small $\rho$ parameters (unbalanced plans that do not sum to one), and removing them increases the error (but remains surprisingly well correlated). Using a simpler output head for the OT plan leads to a failure to predict OT plans.
>
> This explanation and ablation study will be added to the paper.
>
>
> > * D1, D2 input are the same for different GCN and cross attention layers or they are updated? What is the training time/computational complexity of the model?
>
> + D1 and D2 are the inputs to the first GCN and cross attention layer and are indeed not updated along layers. We will clarify this in the text and in Fig. 1
> + The model has worst case (for fully connected graphs) forward computational complexity of $O(n_1n_2+n_1^2+n_2^2)$, but training the model requires computing the FUGW loss for each batch, which is $O(n_1n_2^2+n_2n_1^2)$. So the overall complexity is $O(n_1n_2^2+n_2n_1^2)$. As stated in the Appendix A first paragraph, training the model on the IBC dataset (fMRI) takes about 100 hours on a V100 GPU.
>
> > What is the transferability of the model? For instance, in your simulated experiment, if you train a model with node number in cluster 1, n1, and node number in cluster 2, n2, does the model still work when you change the number in cluster 1 and the number in cluster 2 relatively but keep the total number the same (e.g., n1+2, n2-2)? Or you need to train the model every time you are facing a new environment?
>
> This is a very interesting point. We do not expect our model to be robust to big distribution shifts but it is still relatively robust to small shifts. For the transferability experiment you propose, we test the model trained on balanced SBMs on unbalanced SBMs with cluster ratios sampled following a Dirichlet distribution and report the performances in the table below. We find that even though the relative loss error increases, the Pearson correlation between the ground truth losses and the ULOT losses remains stable.
>
>
> |  | Relative loss error | Pearson correlation |
> | -------- | -------- | -------- |
> | Balanced clusters (train data)    |   $0.19 \pm 0.29$  |  $1$   |
> | Unbalanced clusters (distribution shift) | $0.27 \pm 0.69$    |  $0.98$  |
>
> We will add a figure in the supplemental material showing the error as a function of the distance between the test graph cluster ratios and the uniform ratios to further understand the robustness of the model to data shift.
>
> > \alpha being 0 or 1 are two extreme numbers in the results. What about 0.2 and 0.8? I would like to know whether features or short distance path are the dominant factor in the deep learning model, and how the results different from the ground truth?
>
> During training and testing, for each graph pair the values of $\alpha$ are sampled following a Beta distribution with parameters $(0.5,0.5)$. This implies that the $\alpha$ values span the whole [0,1] range. Consequently, the results of Figure 7 are reported for many different values of $\alpha$ between 0 and 1, and we find that the error is nearly independent of $\alpha$.
>
> To study the impact of the GCN layer, we perform an ablation study of the GCNs on the fMRI dataset. We observe that it drastically decreases the performance from a relative loss error of 19% to 146%, which highlights the fact that the structure information actually helps the neural network find the right solution.
>
> Since our network is differentiable, we plan on adding to the supplemental material a sensitivity analysis of the parts of our model to further investigate the importance of each component, thanks for this very interesting question.
>
>
>
> > The similarity matrix overall had a quite high correlation value (> 0.6). Can this be caused by the sparsity? A diagnal-block high value correlation can be easily affected by the zero connection between different clusters. Also not clear about what the reason of showing fig. 6. MDS (fig 6 right) is not defined in the paper.
>
> The similarity matrix $S$ in Figure 6 shows the transport plan mass $S_{i,j}$ of the ULOT plan between graph $i$ and graph $j$. The values are not correlations but mass, so it makes sense that the values are high, with maximal values at $1$ when all the mass is transported.
>
> We agree that we should add the reference for MDS, which stands for Multidimensional scaling [A] and is a method for visualizing data points in low dimension such that their similarities are preserved. We will add the reference and explanation to the paper.
>
> > An example of transportation plan of real-world data from ULOT and solver will help verify the model, especially if authors can also correlate the results with the biological reasoning.
>
> This is indeed a very interesting question as ULOT can help solve important problems in neuroscience and genomics. Since the loss error is low on fMRI data, we expect the transport plans to be close to the solver transport plans and thus meaningful for real-life applications as was already demonstrated in [30]. We will provide such an illustration in the supplemental material of the final version.
>
> ### References
>
> [A] Kruskal, J. B. (1964). Multidimensional scaling by optimizing goodness of fit to a nonmetric hypothesis. Psychometrika, 29(1), 1-27.
> [30] Alexis Thual, Quang Huy Tran, Tatiana Zemskova, Nicolas Courty, Rémi Flamary, Stanislas Dehaene, and Bertrand Thirion. Aligning individual brains with fused unbalanced gromov wasserstein. Advances in neural information processing systems, 35:21792–21804, 2022.

---

> > ### Comment · Reviewer_HUXF · 2025-08-01
> > **Response**
> >
> > The main revision of the paper should focus on the motivation, idea and logics of the proposed model and its structure. In the current version, simplifying the optimal transport calculation is important, but it is also important to let people understand why and how it works.
> > With additional explanation and experiments shown by the authors, I will increase my rating to 4. Thanks.

---

> > > ### Author Response · Authors · 2025-08-04
> > > **Response**
> > >
> > > We sincerely thank the reviewer for their constructive feedback and their revised rating. In the final version, we will include a detailed explanation of the motivation, intuition, and design choices behind our model, in order to improve the clarity of our contribution.

---

### Official Review · Reviewer_3cZ4 · 2025-06-24

**Clarity:** 4
**Significance:** 4
**Originality:** 4
**Rating:** 6
**Confidence:** 4

**Summary:**

the paper proposes a   deep learning pipeline  (named Unsupervised Learning of Optimal Transport plan prediction (ULOT)) that predicts the  transport plan as a solution to the fused unbalanced Gromov-Wasserstein loss. The proposed pipeline works faster than SOTA solvers, with getting good accuracy. The proposed pipeline can also provide initial guesses to speed up  traditional solvers.

**Questions:**

- line 117: why are $\alpha$ $\beta$ encoded *"at each layer of the network"*? does it mean each layer of the GCN (FIg.1)? why is it not encoded only at the input layer?

 - lines 108+343: can a mix approach (supervised/unsupervised) be used for optimisation?
 e.g. it seems that training sets with G2 being a  sub-graph of G1 (e.g. jackknife/bootstrap resampling) would provide access to ground truth transport plans? would that help with requirement of "large training datasets"?

 - line 257: why is KL used and not cross entropy?

**Ethical Concerns:**

["NO or VERY MINOR ethics concerns only"]

**Final Justification:**

NA

**Limitations:**

yes

**Quality:**

4

**Strengths And Weaknesses:**

**Strengths:**
- the paper is very well presented and clear.
- contributions of the proposed pipeline are well articulated and well validated with experimental results

**Weaknesses:**
$\emptyset$

---

> ### Author Rebuttal · Authors · 2025-07-30
>
> We are grateful for your recognition of the contributions of the paper and hope we can clarify the questions you raised in the following.
>
>
> > line 117: why are alpha and beta encoded "at each layer of the network"? does it mean each layer of the GCN (FIg.1)? why is it not encoded only at the input layer?
>
>
> We designed the model so that the information about $\rho$ and $\alpha$ is not lost along the layers, which is a common issue when models are conditioned on a real value, such as time in diffusion models. Inspired by the method in [A], which has become a standard practice for diffusion models, we concatenate the value of $\rho$ and the encoding for alpha to the node features at the entry of each MLP, GCN and Linear layer.  This design ensures that all layers have direct access to the conditioning parameters and can learn themselves at which layer this information is the most pertinent.
>
> > lines 108+343: can a mix approach (supervised/unsupervised) be used for optimisation? e.g. it seems that training sets with G2 being a sub-graph of G1 (e.g. jackknife/bootstrap resampling) would provide access to ground truth transport plans? would that help with requirement of "large training datasets"?
>
> This is an excellent idea that should be investigated further in future work. The difficulty with the approach is that a ground truth matching (such that the partial matching we would get if one graph is a subgraph of the other) is not always an optimal transport plan. Indeed, for high values of the  $\rho$ parameter, all the mass must be transported, so the ground truth matching in the subgraph setting is no longer the optimal OT plan. This means that we still need to rely on solvers for ground truth plans, which makes training computationally intensive. Also note that all existing solvers only provide a local minimum and training in a supervised way to predict those minimums could lead to a sub-optimal solution.
>
>
> > line 257: why is KL used and not cross entropy?
>
> Given that the ground truth distribution is a one-hot encoding, the KL divergence and the cross entropy are equivalent. We will clarify this in the final version, thanks for your question.
>
>
> ### References
>
> [A] Ho, J., Jain, A., & Abbeel, P. (2020). Denoising diffusion probabilistic models. Advances in neural information processing systems, 33, 6840-6851.

---

### Official Review · Reviewer_evnK · 2025-06-30

**Clarity:** 4
**Significance:** 3
**Originality:** 2
**Rating:** 5
**Confidence:** 3

**Summary:**

In this paper, the authors propose a method for fast approximation of Fused Unbalanced Gromov Wasserstein OT plan with neural nets, trained in an unsupervised way. Their main idea is to directly plug the neural architecture within the FUGW cost, which has been proposed previously for regular OT but, to my knowledge, not for FUGW. The authors argue that one of the main advantage is the computational gain in handling the notoriously difficult quadratic problem associated with GW. Another notable advantage is that everything is differentiable, including with respect to the regularization parameter of the OT problem, or the input graph. Their architecture starts with regular GNNs to compute node representations, then the computation of an OT plan from a mix of cross-attention and fully connected layers. This includes the learning of node weights, as is required by unbalanced OT. The authors then test extensively their architecture on synthetic SBM data, where node features are the community labels. They include (sometimes supervised) optimization with respect to the different hyperparameters, or even the input graph. They also examine real MRI data, where they also test the use of the proposed method as warm-start for other solvers.

**Questions:**

See above

**Ethical Concerns:**

["NO or VERY MINOR ethics concerns only"]

**Final Justification:**

I thank the authors for their answers, and maintain my positive score.

**Limitations:**

Limitations are adressed properly.

**Quality:**

3

**Strengths And Weaknesses:**

Strengths
- the paper is very well-written, clear, and quite complete.
- The proposed method is natural but novel, and handled in a clever way despite being quite non-trivial.
- Experimental results are generally convincing, especially the computational advantages of the proposed method, which is the main strength of the proposed method here

Weaknesses
- I still have one interrogation on the architecture itself, which is not really alleviated by the experiments. It seems the role of node features is rather important compared to the graph structure: the latter only intervene in the early "GNN" part of the architecture, and the computed representations are then buried under several layers of attention, similarities, and so on. This is only an intuition, but node features seem preeminent here. Is the architecture performant when node features are not "useful" ? In the experiments, the node features contain the (noisy) labels, so a great deal of immediate information. The authors do test $\alpha=1$, which would completely ignore the node features when solving FUGW directly, but here the node features are still in the parameterized transport plan, so are not ignored! It would be useful to test the architecture without node features altogether (or with more complex features that noisy labels). In other words, can the proposed architecture handle non-fused GW?
- The authors test a lot of aspects of their architecture, so some small experiments are not necessarily useful, and may take up some space. For instance, the "graph learning" part (differentiability wrt the graph) only tries to learn the identity in the proposed experiments, which is very limited. Maybe it's better to put it in a more developed form in the appendix, or not at all and develop more the other experiments.

---

> ### Author Rebuttal · Authors · 2025-07-30
>
> Thank you for your positive review about our paper. We also appreciate your remarks, as we think they raise fundamental questions regarding the model design.
>
> > I still have one interrogation on the architecture itself, which is not really alleviated by the experiments. It seems the role of node features is rather important compared to the graph structure: the latter only intervene in the early "GNN" part of the architecture, and the computed representations are then buried under several layers of attention, similarities, and so on. This is only an intuition, but node features seem preeminent here. Is the architecture performant when node features are not "useful" ? In the experiments, the node features contain the (noisy) labels, so a great deal of immediate information. The authors do test , which would completely ignore the node features when solving FUGW directly, but here the node features are still in the parameterized transport plan, so are not ignored! It would be useful to test the architecture without node features altogether (or with more complex features that noisy labels). In other words, can the proposed architecture handle non-fused GW?
>
> This is a very interesting question, and you are completely right about the fact that the node features are used even for $\alpha=1$. We think that this is actually a strength of our model, which makes it competitive in situations where the Gromov problem is hard to solve. Indeed, the Gromov problem has many invariants (in many cases, the featureless graphs are invariant to some permutations of nodes), and using the node features breaks these invariants and helps predict a deterministic solution.
>
> In a case where the features are absent or not relevant, one should not use FUGW and ULOT but rather solve pure GW which is a distinct task that needs to be designed differently, probably with stronger graph architectures such as the Evoformer architecture from [B], which has a stronger focus on the graph connectivity matrices. This is an interesting discussion that we will add to the paper.
>
>
> > The authors test a lot of aspects of their architecture, so some small experiments are not necessarily useful, and may take up some space. For instance, the "graph learning" part (differentiability wrt the graph) only tries to learn the identity in the proposed experiments, which is very limited. Maybe it's better to put it in a more developed form in the appendix, or not at all and develop more the other experiments.
>
> We agree that this is not a crucial part of the paper and we will move it to the appendix. The motivation behind it was to demonstrate one use case of the differentiability of the ULOT loss which, to the best of our knowledge, has not been previously explored. It is also a classical problem studied in recent papers about distribution flows to illustrate the trajectory and the geometry of the space [A] on Euclidean distributions and more recently in the GW space [C].
>
>
> ### References
>
>
> [A] Bonet, C., Vauthier, C., & Korba, A. (2025). Flowing Datasets with Wasserstein over Wasserstein Gradient Flows. arXiv preprint arXiv:2506.07534.
>
> [B] Jumper, J., Evans, R., Pritzel, A., Green, T., Figurnov, M., Ronneberger, O., ... & Hassabis, D. (2021). Highly accurate protein structure prediction with AlphaFold. nature, 596(7873), 583-589.
>
> [C] Zhang, Zhengxin, Ziv Goldfeld, Kristjan Greenewald, Youssef Mroueh, and Bharath K. Sriperumbudur. "Gradient Flows and Riemannian Structure in the Gromov-Wasserstein Geometry." arXiv preprint arXiv:2407.11800 (2024).

---

### Official Review · Reviewer_bmor · 2025-07-05

**Clarity:** 2
**Significance:** 2
**Originality:** 3
**Rating:** 4
**Confidence:** 4

**Summary:**

The paper proposes a deep learning method called Unbalanced Learning of Optimal Transport (ULOT) for predicting optimal transport plans between unbalanced graphs, addressing the computational inefficiency of classical solvers. The method minimizes the Fused Unbalanced Gromov-Wasserstein loss using a novel neural network architecture that integrates cross-attention and graph neural networks. This architecture is conditioned on the FUGW hyperparameters, allowing ULOT to generalize across different parameter values.

**Questions:**

See weaknesses.

**Ethical Concerns:**

["NO or VERY MINOR ethics concerns only"]

**Final Justification:**

The rebuttal clarified the generalization experiments and somehow addressed my concerns. So I raised the rating to 4.

**Limitations:**

Yes.

**Paper Formatting Concerns:**

No.

**Quality:**

3

**Strengths And Weaknesses:**

Strengths:
1.	Using neural networks to approximate the transportation plan seems a new task and the inference time can be much less than traditional solvers.
2.	The paper is well organized. The method is relatively simple, but the experiments are comprehensive.
3. Results are quite good. It is easy to understand ULOT has better speed but it is surprising to see it even has lower error than other solvers.



Weaknesses:
1. The motivation of the model design is somehow not very clear. Why does the designed neural network can approximate the transportation plan? how well can it approximate the transportation plan (is there any theoretic bound)?
2. Some training details are also lacking, e.g. what solvers are used as ground truth and what is the loss function?
3. I assume one of the major drawbacks of using neural network for OT approximation is generalization. It would be good to see if the method can perform well when the test data is very different from training data.

---

> ### Author Rebuttal · Authors · 2025-07-30
>
> Thank you for accurately summarizing our paper and providing a fair evaluation of its strengths and weaknesses. We appreciate your questions and have done our best to answer them below.
>
>  > Results are quite good. It is easy to understand ULOT has better speed but it is surprising to see it even has lower error than other solvers.
>
> We would like to clarify why the ULOT loss is sometimes lower than the solver loss, which is explained by two factors.
>
> First, ULOT can outperform solvers when they converge to poor local minima. This happens for instance in scenarios such as the one presented in Figure 3 (left) where the constant plan equal to zero is a local minima. Because of its design that implies continuity with respect to its input, ULOT often avoids these local minima, which we believe is a crucial asset.
>
> Second, in cases such as the one depicted in Figure 7 (right), ULOT has lower error than the entropic solver with high regularization, which is expected. Indeed, the entropic regularization is introduced to speed up the optimization at the cost of some non-optimality in the transport plan. The intuition is that lower entropic regularization leads to lower error but slower convergence.
>
>
> > The motivation of the model design is somehow not very clear. Why does the designed neural network can approximate the transportation plan?
>
> We appreciate your question, as it is fundamental that the model design be clearly motivated and we will improve this aspect in the final version. We designed the model so that it can learn transport plans that respect the conditions for minimizing the FUGW loss (unbalanceness, feature and structure integration). We originally tried other simpler architectures, but their limited performance led to the current architecture that we detail below.
>
> * The first block outputs node features that are relevant in a matching context, and is an adaptation of [13]. Intuitively the cross attention mechanism computes node features that are similar for nodes that can be matched and dissimilar for nodes that should not. The use of GCNs is motivated by the Gromov term in the FUGW loss that enforces that the model should take into account the graph topology (features are already used in MLP and cross-attention).
> * The optimal transport plan prediction block is designed so as to predict an unbalanced OT plan that must contain pairwise relationships between nodes but also that can discard (reweight) nodes that cost too much from an OT perspective. This is done by separating the two aspects:
>     *  We chose to use for the pairwise relationship in the OT plan the average of the cross attention matrix to which we apply a row-wise softmax and a column-wise softmax respectively.
>     * The pairwise matrix above gives a good starting point for the plan, which is then refined in the unbalanced optimal transport plan block, where the scaling per node makes it no longer balanced. This is why we also provide in the last block two node reweighting heads (Linear+sigmoid and multiplication of the plans by the diagonal weight matrices) that allow scaling on the OT plan and enable the network to discard mass.
>
> We provide an ablation study of the impact of each step of the OT plan block on the SBM dataset in the table below. As a first ablation we remove the node scaling in the OT plan head, as a second ablation we replace the whole OT plan head by a much simpler nonlinear transformation of the node embedings to recover an OT plan with  ($P_\theta^{(\alpha, \rho)})_{i,j} = \frac{1}{1+||(F^{\text{final}}_1)_i -(F^{\text{final}}_2)_j||^2_2},$ where $F^{\text{final}}_1$ and $F^{\text{final}}_2$ are the outputs of the GCN and cross attention block for the first and second graphs respectively.
>
>
> | Ablation |   Relative error   |  Pearson correlation |
> | -------- | -------- | -------- |
> |  None   |   $0.19 \pm 0.29$  |  $1.0$ |
> | Remove node scaling | $4.1 \pm 9.5$    |  $0.89$  |
> | Replace OT plan block by nonlinearity | $27 \pm 49$   |  $0.48$  |
>
> We can see that node scaling is essential for recovering unbalanced OT plans especially for small $\rho$ parameters (unbalanced plans that do not sum to one), and removing them increases the error (but remains surprisingly well correlated). Using a simpler output head for the OT plan leads to a failure to predict OT plans.
>
> This explanation and ablation study will be added to the paper.
>
>
> > how well can it approximate the transportation plan (is there any theoretic bound)?
>
> We provide empirical results about the quality of the OT plans and of their approximation. To the best of our knowledge there is no theoretical guarantee about the estimation quality of amortized optimization [3], so this is still an open question.
>
> > Some training details are also lacking, e.g. what solvers are used as ground truth and what is the loss function?
>
> The loss function is the FUGW loss and is introduced in equation (4). The solvers used as ground truth are presented in subsection 2.1. We used the IBPP [39] solver for all the results except for Figure 7 (right), where we compare many different solvers, namely the MM [6], LBFGSB [30] and Sinkhorn solvers [8]. When using several numerical solvers, the ground truth is taken as the solution with the smallest objective.
>
> We will provide in the final version more details about all those solvers in the supplementary of the paper to be more self-contained.
>
> > I assume one of the major drawbacks of using neural network for OT approximation is generalization. It would be good to see if the method can perform well when the test data is very different from training data.
>
> This is a very interesting point that is crucial in understanding the potential of ULOT for applications. We would like to clarify that all the performances reported are generalizations on test data, i.e. graph pairs that were not used for training, which testifies for the ability of the model to generalize to different pairs within the same distribution.
>
> The question of robustness to data shift is indeed important. We tested the following distribution shift: we trained ULOT on a dataset of Stochastic Block models with balanced clusters (every cluster has the same number of nodes), and tested the model on a dataset of SBMs with unbalanced clusters, where the cluster ratio is sampled following a Dirichlet distribution. We report the results in the table below and find that even though the relative loss error increases, the Pearson correlation between the ground truth losses and the ULOT losses remains stable.
>
>
> |  | Relative loss error | Pearson correlation |
> | -------- | -------- | -------- |
> | Balanced clusters (train data)    |   $0.19 \pm 0.29$  |  $1$   |
> | Unbalanced clusters (distribution shift) | $0.27 \pm 0.69$    |  $0.98$  |
>
>
> ### References
>
> [6] Laetitia Chapel, Rémi Flamary, Haoran Wu, Cédric Févotte, and Gilles Gasso. Unbalanced opti- mal transport through non-negative penalized linear regression. Advances in Neural Information Processing Systems, 34:23270–23282, 2021.
>
> [39]Yujia Xie, Xiangfeng Wang, Ruijia Wang, and Hongyuan Zha. A fast proximal point method for computing exact wasserstein distance. In Uncertainty in artificial intelligence, pages 433–453. PMLR, 2020.
>
> [8] Lenaic Chizat, Gabriel Peyré, Bernhard Schmitzer, and François-Xavier Vialard. Unbalanced optimal transport: Dynamic and kantorovich formulations. Journal of Functional Analysis, 274(11):3090–3123, 2018.
>
> [30] Alexis Thual, Quang Huy Tran, Tatiana Zemskova, Nicolas Courty, Rémi Flamary, Stanislas Dehaene, and Bertrand Thirion. Aligning individual brains with fused unbalanced gromov wasserstein. Advances in neural information processing systems, 35:21792–21804, 2022.
>
> [13] Yujia Li, Chenjie Gu, Thomas Dullien, Oriol Vinyals, and Pushmeet Kohli. Graph matching networks for learning the similarity of graph structured objects. In International conference on machine learning, pages 3835–3845. PMLR, 2019.
>
> [3] Amos, B. (2023). Tutorial on amortized optimization. Foundations and Trends® in Machine Learning, 16(5), 592-732.

---

### Decision · Program_Chairs · 2025-09-17

**Decision:**

Accept (poster)

**Comment:**

This paper introduce ULOT, a deep learning method that predicts optimal transport plans between graphs using a cross-attention architecture trained on FUGW loss. It is scalable, differentiable, and allows graph alignment with applications in synthetic and brain network. The reviewers were already quite positive about this work before rebuttal, and the discussion led to a consensus on the qualities of this submission. Hence, I recommend acceptance to NeurIPS 2025.